# Influence of Experiencing Bullying Victimization on Suicidal Ideation and Behaviors in Korean Adolescents

**DOI:** 10.3390/ijerph182010853

**Published:** 2021-10-15

**Authors:** Jiyoun Kim, Young Ko

**Affiliations:** 1Department of Nursing, College of Nursing, Kyungdong University, Wonju 26495, Korea; iscraa08@kduniv.ac.kr; 2Department of Nursing, College of Nursing, Gachon University, Incheon 21936, Korea

**Keywords:** suicide, bullying, adolescent, depression, anxiety, social support, victim

## Abstract

This study was conducted to identify the association between experiencing bullying victimization and suicidal ideation and behaviors, so as to identify the influencing factors on suicidal ideation, suicide plans, and suicide attempts among Korean adolescents. This study analyzed data from the Korean Psychosocial Anxiety Survey. The survey used nationally representative samples of Korean adolescents aged 14 to 18 years old. The experience of bullying victimization negatively influenced suicidal ideation and suicide plans, but not suicide attempts. This influence was reduced, but still remained after controlling depression, anxiety, relationships with family, relationships with teachers and friends, and social support. Influencing factors differed with suicidal ideation, plans, and attempts. Anxiety was the only factor associated with all stages of suicidality. When developing future interventions, health providers need to consider the differences in the factors associated with each stage of suicidality in adolescents.

## 1. Introduction

Suicide is a significant health problem worldwide, and was the leading cause of death among Korean adolescents aged 10–19 years in 2018 [1]. The suicide rate among South Koreans aged 10 to 24 years was 7.6 people per 100,000, which is higher than the Organization for Economic Cooperation and Development (OECD) average of 6.5 people per 100,000 [2]. The survey on the Rights of the Children and Youth in South Korea showed that 33.8% of students aged 10 to 19 years considered suicide due to stress; anxiety; conflicts with family, seniors, and their peers; and economic difficulties [3]. In the OECD Program for International Student Assessment, Korean students performed relatively high in academic achievement, but their life satisfaction was below the OECD average [4]. Therefore, attention should be paid to their life situation and difficulties in the home and school in order to solve social issues related to child and youth suicide.

Many studies have reported that suicidal ideation, plans, and attempts are highly correlated [5]. However, in an ideation to action framework, the development of suicidal ideation and the transition to a suicide attempt are considered distinct processes [5]. Several studies support this framework, reporting differences in the factors that influence the development of suicidal ideation and progress from ideation to attempts [6,7]. For example, Wetherall and colleagues [6] reported that they found that only volitional phase variables such as acquired capability, mental imagery about death, impulsivity, and being more likely to know a friend who had made a suicide attempt differentiated between the suicide attempt group and the suicidal ideation group among young adults. Mar and colleagues [7] reported that adolescence who attempted suicide were more likely to report exposure to self-harm in others and have a psychiatric disorder such as depression, anxiety disorder, and behavioral disorder compared with those who had experienced suicidal ideation. However, many have not been examined within an ideation to action framework before. Therefore, it is important to identify the factors that differentiate those who attempt suicide from those who exhibit suicidal ideation or plans but do not act on these thoughts or plans.

One of the possible influencing factors on suicide related behaviors is experiencing bullying victimization [8]. Bullying is defined broadly as the execution of a harmful action characterized by intention, repetition, and either a physical or a psychological power imbalance [9]. In particular, the increasing perception that physical violence can be subject to legal and social penalties in Korea has led to an increase in psychological violence such as bullying, verbal violence, and cyberbullying. The reported prevalence of school bullying increased from 13.3% in 2012 to 23.2% in 2019 [10].

Bullying, by its nature, is likely to have certain characteristics (e.g., fear of reporting it by the victim) and certain outcomes (e.g., the development of low self-esteem and depression in the victim) [9]. Victims are adversely affected by the mental stress associated with situations that they cannot avoid, and they also feel isolated from their peer groups. The resulting tremendous psychological distress that the victim can feel internally can manifest as internalizing behaviors, such as depression or suicidal ideation [11]. It can lead to externalizing behaviors, such as suicide attempts or physical self-harm [11,12].

The deleterious effects of bullying on victims are of great concern. Adolescence, with an increase in emotionality, risk-taking, and focusing more on peer relationship, is an at-risk period for the development of suicidality following exposure to bullying [13]. Several studies have shown the significant relationship between the experience of bullying victimization and suicidality among adolescents [14,15,16,17,18,19,20,21,22]. However, little research has been conducted on whether the experience of bullying victimization influences the transition of suicidal ideation to action [23,24,25,26]. Moreover, these studies have reported inconsistent results. Hinduja and Patchin [23] reported that students who reported being bullied at school and online were more likely to report suicidal thoughts and attempts. However, Holt and colleagues [24] reported that bullying involvement is associated with an increased risk of suicidality, and being a bully-victim is associated with the greatest risk of suicidal ideation.

The experience of bullying victimization, along with various factors such as demographic characteristics, socioeconomic status, mental health status, and poor social support, may increase adolescent suicide risk. For example, the association between the experience of being bullied and suicidal ideation and attempts was slightly stronger in girls than in boys [18,19,22]. Older girls who experienced bullying had a higher risk of suicidal ideation than younger girls or boys [17]. It has been reported that family support or social support moderated the relationship between victimization and suicidal ideation in several studies [16,20,21]. Bullying victims were more likely to experience anxiety symptoms and depression, both of which are risk factors for suicidal ideation [26,27]. Experience of bullying has been found to induce suicidal thoughts and attempts, in part through mental health such as depression or aggression [15,28]. Therefore, it is necessary to consider various factors together when examining the relationship between bullying and suicidal-related behaviors.

Previous studies examining the relationship between bullying and suicide-related behavior have several limitations, in that they do not research national in scope or do not consider various factors related to the experience of bullying and suicidality [14,15,17,22,23,25]. The Korean Psychosocial Anxiety Survey conducted in 2015 is the only large-scale survey to investigate psychosocial anxiety through a representative sample of Korean adolescents [29]. This survey includes not only individual characteristics, but also various psychological and social factors such as depression and anxiety related to experiences of bullying and suicide-related behaviors [29]. In this study, the relationship between experiences of bullying and suicide-related behaviors was investigated using data from the Korean Psychosocial Anxiety Survey.

The purpose of this study was to identify (1) an association between the experience of bullying victimization and suicidal ideation and behaviors (Aim 1), and (2) factors the development of suicidal ideation and transition from suicidal ideation to action among Korean adolescents focused on the experience of bullying victimization, mental health, and relationship with family, relationships with teachers and friends, and social support (Aim 2).

## 2. Materials and Methods

### 2.1. Study Design and Participants

This study analyzed the data from the Korean Psychosocial Anxiety Survey, which was a population-based representative sample of 5000 adolescents aged 14 to 18 years and 7000 adults aged at least 19 years. The survey was conducted by the Korean Institute of Health and Social Affairs (KIHASA, Sejoing, South Korea) in 2015 [29]. This survey was designed to identify the status of psychosocial anxiety and its key influencing factors [29]. For this study, the study population was limited to adolescents aged 14 to 18 years. The adolescent participants were selected using multistage stratified probability sampling based on the geographical area, gender, and age to ensure that they were a representative sample of the entire country. We used the data from 5000 participants initially selected in this analysis.

The sample size was calculated to address Aim 2. The required sample size for the multiple logistic analysis was calculated based on the recommendations of Peduzzi et al. [30] as follows. Let *P* be the smallest proportion of cases in the population, and k be the number of independent variables in logistic analysis. In this study, adolescents with an experience of being bullied comprised 16.2% of suicide ideators, 15.9% of suicide planners among suicide ideators, and 17.1% suicide attempters among suicide planners. The minimum sample size was *N* = 10 × k/*P* [30], and so with *P* = 15.9% and k = 10 in this study, 629 participants were required. Therefore, the predetermined sample size was sufficient to achieve Aim 2.

### 2.2. Measure

#### 2.2.1. Experience of Bullying Victimization

Experience of bullying victimization was measured using the following self-reported question: “Have you ever been bullied by classmates or peers in life-time?”. The following responses were allowed: “In the past and now” (1 point), “In the past but not at present” (2 points), “Not in the past but now” (3 points), or “Not in the past nor at present” (4 points). In this study, the cases in which any bullying had been experienced in the past or at present were classified as “experience of bullying victimization”, and those with no experience of bullying victimization from the past to the present were classified as “no experience of bullying victimization”.

#### 2.2.2. Suicidality: Suicidal Ideation, Suicide Plans, and Attempts

The following suicidality-related measures were measured using self-reported questions: suicidal ideation, suicide plans, and attempts. Suicidal ideation was assessed using the following question: “Have you thought about ending your life through any method in the past year?”. If suicidal ideation was present, the following question on suicide plans was asked: “Have you ever planned to attempt suicide specifically in the past year?”. If a suicide plan was present, the following question on suicide attempts was asked: “Have you ever attempted suicide in the past year?”.

For this study, participants who reported no suicidal ideation were included in the “no suicidal ideation” group. Those who reported suicidal ideation, but no suicide plans, were included in the “suicidal ideation only” group. Those who thought about and planned suicide but did not attempt suicide were included in the “suicide plan” group. Those who thought about, planned, and attempted suicide were included in the “suicide attempt” group.

#### 2.2.3. Depression

The Center for Epidemiological Studies Depression scale (CES-D) is a self-reported screening tool for depressive symptoms developed for epidemiological studies by the National Institute of Mental Health, and it consists of 20 items covering four factors [31]. The respondents rated how frequently each item applies to them during the previous week on a four-point Likert scale ranging from 0 (rarely or none of the time) to 3 (most or all of the time). The four-factor structure covers the depressive affect, absence of a positive effect or anhedonia, somatic activity or inactivity, and interpersonal challenges [31]. CES-D was designed to identify the high-risk groups with a cutoff score of 16 used in the 20-item version [31].

The present study evaluated depressive symptoms using the four-factor, 11-item version of the scale (CES-D11), whose construct validity and reliability were tested and confirmed [32]. CES-D11 was translated into Korean, and the validity and reliability of that version were also tested [33]. They reported that while the four factors of CES-D11 did not fit the Korean population, the total CES-D11 score was significant at identifying the high-risk groups. The total score corresponding to the summation of all item scores in the CES-D11 was multiplied by 20/11, with a CES-D11 score higher than 8.8 (=16/20 × 11), indicating the presence of depressive symptoms [34]. Cronbach’s α was 0.81 for CES-D11 compared to 0.86 for CES-D [32]. The present study also found that the reliability of CES-D11 was good (Cronbach’s α = 0.86).

#### 2.2.4. Anxiety

Anxiety was measured using the Zung Self-rating Anxiety Scale (SAS) [35], which consisted of 20 items (five on affective symptoms and 15 on somatic symptoms) scored on a four-point Likert scale from 1 (never) to 4 (always). The total SAS score ranged from 20 to 80, with a higher score indicating greater anxiety. A cutoff score of 45 was used [35] to divide the participants into two groups: those with and those without anxiety. Lee [36] confirmed the convergent validity and test−retest reliability for the Korean population. In the present study, the reliability of the SAS was good (Cronbach’s α = 0.80).

#### 2.2.5. Relationships with Family and Teachers and Friends, and Subjective Social Support

The family relationship was assessed using the following question: “How is your relationship with your family?”. The responses to this question were quantified using a four-point Likert scale ranging from very bad (1 point) to very good (4 points). The response was classified as “bad” (very bad and bad) and “good” (very good and good).

The school relationships with teachers and friends were assessed through the following question: “How are your relationships with your teachers and friends?”. The responses to this question were also quantified using a four-point Likert scale ranging from very bad (1 point) to very good (4 points). The response was classified as “bad” (very bad and bad) and “good” (very good and good).

The subjective social support was assessed using the following question: “How much social support have you received?”. The responses were rated from 0 points (none) to 10 points (a lot), with scores between 0 and 4 points classified as low subjective social support.

#### 2.2.6. Sociodemographic Characteristics

Sociodemographic characteristics, including gender, age, subjective family class, and living arrangement, were assessed using a structured questionnaire. The subjective family class was measured using a five-point Likert scale ranging from 1 (lowest class) to 5 (highest class). The bottom two classes were classified as “low”, and the top three were classified as “not low”. The living arrangement was classified into “living with parents” and “living with a single parent or without parents”.

### 2.3. Data Collection

The data for the Korean Psychosocial Anxiety Survey were collected using a Web-based survey [29]. E-mail requests were sent to the potential participants. They were required to read and sign a consent form before participating via an online survey and they agreed to be available for further research. The researcher was provided with de-identified data from the KIHASA after they approved the request. The Investigational Review Board of the university to which the researchers were affiliated approved the protocol for the secondary analysis using these data (IRB No. 1044396-202002-HR-057-01).

### 2.4. Statistical Analysis

We used the SPSS 25.0 software program for this analysis. Descriptive statistics were used to identify the characteristics of the sociodemographic characteristics, depression, anxiety, relationships with family and school, subjective social support, an experience of bullying victimization, and suicidality (suicidal ideation, suicide plans, and attempts) among participants.

The differences in the sociodemographic characteristics, depression, anxiety, relationships with family, relationships with teachers and friends, and subjective social support, and suicidality according to the experience of bullying victimization were identified using a Chi-square test. Hierarchical logistic regression analyses were used to identify the factors influencing suicidality, namely: suicidal ideation, suicide plans, and suicide attempts (Aims 1 and 2). To order to identify which variables independently distinguished the groups, multiple hierarchical logistic regressions were performed: one with the “no suicidal ideation” group as the reference, another with the “suicidal ideation only” group as the reference, and the other with the “suicide plan” group as the reference.

The hierarchical logistic analyses were implemented as follows. Sociodemographic characteristics, and experience of bullying victimization, depression, and anxiety were entered into the first model, and the relationships with family and school, and subjective social support were entered into the second model. Odds ratios (OR) indicated the likelihood of membership in the “suicidal ideation only” group (relative to the “no suicidal ideation” group, the “suicide plan” group (relative to the “suicidal ideation only” group), and the “suicidal attempt” group (relative to the “suicide plan” group). The Hosmer−Lemeshow goodness-of-fit index was used to assess how well the model fitted the data, and Nagelkerke’s R^2^ was used to examine the explanatory power of the model. *p* values < 0.05 were considered significant.

## 3. Results

### 3.1. Experience of Bullying Victimization and Suicidality

The characteristics of the participants are shown in Table 1; most were male (52.2%), and females experienced more bullying victimization than males. The rate of experience of bullying victimization increased with age (χ^2^ = 46.99, *p* < 0.001). Those with more bullying victimization had more severe depression (χ^2^ = 202.70, *p* < 0.001) and anxiety (χ^2^ = 208.13, *p* < 0.001). On the other hand, those with less experience of bullying victimization had a higher subjective social support (χ^2^ = 94.29, *p* < 0.001), a better relationship with family (χ^2^ = 42.92, *p* < 0.001), and better relationships with teachers and friends (χ^2^ = 141.30, *p* < 0.001).

In the overall sample of 5000 participants, 3316 (66.3%) reported no history of suicidal thoughts, plans, or attempts (“no suicidal ideation” group); 1264 (25.3%) reported suicidal thoughts (“suicidal ideation only” group), but no suicide plans and attempts; 219 (4.4%) reported suicide plans, but no suicide attempts (“suicide plan” group); and 201 (4.0%) reported suicide attempts (“suicide attempt” group).

### 3.2. Factors Influencing Suicidal Ideation, Plans, and Attempt

Before deciding on the final analysis structure, this study examined whether there were significant interactions among the independent variables: interactions between the experience of bullying victimization and variables related to relationships with family and school, and subjective social support. Entering all of the interactions into Model 2 yielded no statistically significant interactions. Hence, the following models were selected.

Table 2 lists the results of the hierarchical logistic regression analysis aimed at identifying the factors influencing the suicidal ideation in Model 1, experience of bullying victimization (OR = 1.67 CI = 1.45–1.91), depression (OR = 3.35, 95% CI = 2.82–4.98), and anxiety (OR = 1.98, 95% CI = 1.68–2.33) significantly influenced the development of suicidal ideation. In Model 2, the variables related to relationships with family, relationships with teachers and friends, and subjective social support were entered as protective factors. The subjective social support (OR = 1.79, 95% CI = 1.52–2.20) and the relationship with family (OR = 1.42, 95% CI = 1.14–1.75) influenced the development of suicidal ideation significantly. Entering those variables decreased the influence of depression (OR = 3.52 → OR = 2.88) and anxiety (OR = 1.98 → OR = 1.87), as well as the experience of bullying victimization (OR = 1.67 → OR = 1.58) on the development of suicidal ideation.

Table 3 lists the results of the hierarchical logistic regression analysis aimed at identifying the factors influencing the suicide plans. In Model 1, the experience of bullying victimization (OR = 1.90, 95% CI = 1.50–2.40), depression (OR = 1.91, 95% CI = 1.46–2.50), and anxiety (OR = 1.47, 95% CI = 1.12–1.91) had a significant influence on the transition from suicidal ideation to plans. In Model 2, the relationship with family (OR = 1.96, 95% CI = 1.46–2.63) had a significant influence on the transition from suicidal ideation to plans. On the other hand, subjective social support (OR = 1.26, 95% CI = 0.98–1.63) and relationships with teachers and friends (OR = 0.75, 95% CI = 0.53~1.10) did not influence the transition from suicidal ideation to plans significantly. Entering the protective factors into Model 2 decreased the negative influences of depression (OR = 1.91 → OR = 1.77), anxiety (OR = 1.47 → OR = 1.42), and the experience of bullying victimization (OR = 1.90 → OR = 1.89) on the transition from suicidal ideation to plans.

Table 4 lists the results of the hierarchical logistic regression analysis aimed at identifying influencing the suicide attempts. In model 1, only anxiety (OR = 1.84, 95% CI = 1.15–2.97) was a significant factor influencing transition from suicide plans to attempts. Entering the protective factors into model 2 decreased the negative influence of anxiety on transition from suicide plans to attempts (OR = 1.84 → OR = 1.77).

## 4. Discussion

This study was conducted to explain the relationship between experiencing bullying victimization and suicidality, and to identify the factors in the development of suicidal ideation and the transition from suicidal ideation to action among Korean adolescents.

Experience of bullying victimization was significantly related to the development of suicidal ideation and the transition from suicidal ideation to plans, but not to transition from suicide plans to attempts. Bullying victimization, as an experience of violence, can be expected to increase the possibility of suicidal attempts [8]. However, these results in this study showed that factors other than experiencing bullying victimization influence the transition from suicide plans to attempts. This is similar to previous results that psychiatric disorders such as depression, anxiety disorder, and behavioral disorder influence the transition from suicidal ideation to attempts, rather than an experience of being bullied [25].

Previous studies examining the relationship between bullying victimization and suicide attempts produced inconsistent findings [22,23,24,25]. One plausible reason for these inconsistent results may be the difference in the type and definition of bullying among studies. Previous studies have shown that the experience of physical bullying was more closely related to suicidal ideation, planning, and attempts than verbal-relational bullying [37,38]. Physical bullying is legally restricted in Korea, so it is possible that the experience of non-physical bullying may have a weaker relevance to suicidal attempts in this study. Nevertheless, we found that the experience of bullying victimization is a risk factor that increases suicidal thoughts and plans. Therefore, it is necessary to give special attention and support to adolescents who have experienced bullying victimization in order to reduce suicidal thoughts or suicide plans.

The present findings highlight the need for different approaches for the stages of suicidality among adolescents. In this study, depression and anxiety were highly related to experiences of bullying and were also related to suicidal thoughts and suicide plans. This result is consistent with those of previous studies [26]. These results imply that mental health may mediate some of the influence of the experiences of bullying victimization on suicidal ideation and plans [15,28]. Accordingly, screening and applying interventions for mental health problems, such as depression and anxiety due to victimization of school bullying in adolescents, are recommended to avoid subsequent internalizing problems.

In this study, experiences of bullying victimization and anxiety were related, but anxiety had more of an influence than an experience of bullying victimization on the transition from suicide plans to attempts. As a result of a review, consistent evidence for a significant difference between anxiety and suicide-related behaviors is reported [27]. This study is similar to a previous study, where there was an association between verbal victimization and suicidal attempts among adolescents with anxiety who perceived low parental support [15]. Recent research reported that impulsiveness, severe anxiety, panic attacks, and agitation along with depression are often immediate suicide risk factors [39]. The interaction of anxiety and other factors, including experiencing bullying victimization or depression, may influence suicide attempts. However, few studies have examined the causal relationship between anxiety and suicide attempts. Therefore, it is necessary to reconfirm these results in studies with a rigorous research design.

In this study, we found that helping ease the anxiety of adolescents might be effective at reducing the suicide rate. More than 60% of Korean adolescents reportedly experience great anxiety about their academic achievement, entrance to university, and their future in general [29]. The increasing rate of suicides due to academic dissatisfaction resulted in many of the Korean adolescents having friends who had attempted suicide [3]. Volitional phase risk factors, such as being more likely to know a friend who had attempted suicide and experienced violence [6], may increase the likelihood of adolescents attempting suicide among those with suicidal ideation or plans. Therefore, there is an urgent need to provide social measures to reduce anxiety about their futures.

The relationships with family and social support reduced the negative influence of the experience of bullying victimization on suicidal ideation. A more positive relationship with their family helped with reducing the negative influence of an experience of bullying victimization on the transition from suicidal ideation to plans. This result was consistent with those of previous studies [16,20]. Family characteristics, such as living with parents and perceiving their own family as being of a high class, were found to be protective factors against bullying and suicide-related behaviors in this study. Similar results were found in other studies [40,41].

The relationship with teachers or friends did not have a protective role in suicidal ideation, suicide plans, and suicide attempts in Korea. These findings are inconsistent with those of studies conducted in other countries [16,21,42,43]. Those inconsistencies may be due to the different school cultures and school systems. Our study findings imply that there are issues in school life focused on academic achievement and competition among students in Korea. Kim and Shin [42] reported that teachers are evaluated based mainly on education outcomes, which is problematic because they make teachers less able to understand and communicate with their students. Existing research showing that the relationship between bullying and suicide is important not only at an individual level, but also in the context of the school, support the importance of the role of the school [23,44]. Therefore, it is necessary to strengthen the support networks in school, especially for adolescents with poor family support.

This study had some limitations. First, the study was a cross-sectional design, and so it could not confirm the causal relationship between the related factors and suicide-related behaviors. Second, the timing and type of bullying experience were unclear, and so the negative consequences of an experience of victimization may have been either underestimated or overestimated. Potential recall bias about the experience of bullying victimization was present. Lastly, as a secondary analysis, this study did not include factors such as exposure to the history of psychiatric disorder, impulsivity, prior self-injurious behaviors, and exposure to self-harm behaviors [45], which may influence the association between the experience of bullying victimization and suicidality.

## 5. Conclusions

Bullying and suicide are significant issues for adolescents and society as a whole. This study found that experiencing bullying victimization influenced the development of suicidal ideation and the transition from suicidal ideation to plans significantly. Anxiety was a factor associated with all stages of suicidal ideation, suicide plans, and suicide attempts. Based on the results of this study, we propose the following implications. First, the support system and level of depression and anxiety need to be assessed before providing counseling services to adolescents suffering from bullying. Those with poor mental health and weak relationships with family and society constitute a group at a high risk of suicide, and so they need intensive care. Psychological interventions that address anxiety should be provided to them to prevent suicide attempts. Second, we found the differences in the factors influencing suicidal ideation and actions. When developing future interventions, school nurses, teachers, and health providers need to consider the differences in the the factors associated with each stage of suicidality in adolescents. It is necessary to strengthen the supporting system of schools and communities for the mental health of adolescents in Korea. Fourth, experiences of bullying can worsen adolescents’ mental health, leading to suicidal thoughts and plans. Therefore, family members, teachers, school nurses, and the community should work toward reducing school bullying. Efforts to reduce bullying in schools should be given priority, and raising students’ ethical awareness and improving their social skill can be done through various school activities. Finally, future long-term studies on mental health and suicidality are needed among adolescents experiencing bullying victimization or perpetration.

## Figures and Tables

**Table 1 ijerph-18-10853-t001:** Comparison of participants’ characteristics by experience of bullying victimization (*N* = 5000).

Variables	Category (Range)	Experience of Bullying Victimization			Χ^2^ or t (*p*)
		Total	No	Yes	
		n or M ± SD	n (%) or M ± SD	n (%) or M ± SD	
**Total**		5000 (100.0)	1682 (33.6)	3318 (66.4)	
Gender	Male	2611 (52.2)	671 (39.9)	1940 (60.1)	154.37 (<0.001)
	Female	2389 (47.8)	1011 (42.3)	1378 (57.7)	
Age (years)	14	741 (14.8)	180 (24.3)	561 (75.7)	46.99 (<0.001)
	15	1192 (23.8)	372 (31.2)	820 (68.8)	
	16	818 (16.4)	293 (35.8)	385 (64.2)	
	17	1045 (20.9)	384 (36.7)	661 (63.3)	
	18	1204 (24.1)	453 (37.6)	751 (62.4)	
Subjective family class	Low (1–2)	1330 (26.6)	569 (42.8)	761 (57.2)	67.84 (<0.001)
	Not low (3–5)	3670 (73.4)	1113 (30.3)	2557 (69.7)	
Living with Parent	Both	4034 (80.7)	1306 (32.4)	2728 (67.6)	99.43 (<0.001)
	Single or non	966 (19.3)	376 (38.9)	590 (61.1)	
Depression	Normal (0–15)	3940 (78.8)	1131 (28.7)	2809 (71.3)	202.70 (<0.001)
	High (≥16)	1060 (21.2)	551 (52.0)	509 (48.0)	
Anxiety	Normal (≤44)	3042 (60.8)	848 (27.9)	2194 (72.1)	208.13 (<0.001)
	High (45~80)	1958 (39.2)	834 (42.6)	1124 (57.4)	
Subjective social support	Low (≤5)	1070 (21.4)	493 (46.1)	577 (53.9)	94.29 (<0.001)
	High (6~10)	3930 (78.6)	1189 (30.3)	2741 (69.7)	
Family relationship	Bad	550 (11.0)	253 (46.0)	297 (54.0)	42.92 (<0.001)
	Good	4450 (89.0)	1429 (32.1)	3011 (67.9)	
School relationship (teachers, friends)	Bad	382 (7.6)	234 (61.3)	148 (38.7)	141.30 (<0.001)
	Good	4618 (92.4)	1448 (31.4)	3170 (68.6)	
Suicidality	No suicidal ideation	3316 (66.3)	886 (26.8)	2430 (73.2)	264.31 (<0.001)
	Suicidal ideation only	1264 (25.3)	537 (42.5)	727 (57.5)	
	Suicide plan	219 (4.4)	130 (59.4)	89 (40.6)	
	Suicide attempts	201 (4.0)	129 (64.2)	72 (35.8)	

**Table 2 ijerph-18-10853-t002:** Factors associated with suicidal ideation.

Variables (Reference)	Suicidal Ideation Only (vs. No Suicidal Ideation ^a^)	
	Model 1 Adjusted OR (95% CI)	Model 2 Adjusted OR (95% CI)
Male (female)	1.89 ** (1.66~2.14)	2.01 ** (1.76~2.30)
Age	0.94 (0.89~0.97)	0.93 (0.89~0.98)
Living with single or non-parent (living with parents)	1.01 (0.84~1.22)	1.00 (0.86~1.26)
Low subjective family class (not low)	1.79 ** (1.55~2.06)	1.47 * (1.26~1.71)
Bullying victimization (no experience)	1.67 ** (1.45~1.91)	1.58 ** (1.38~1.82)
Depressive symptoms (no symptoms)	3.35 ** (2.82~3.98)	2.88 ** (2.41~3.44)
Anxiety (no anxiety)	1.98 ** (1.68~2.33)	1.87 ** (1.59~2.21)
Low subjective social support (not low)		1.79 ** (1.52~2.10)
Bad family relationship (good)		1.42 * (1.14~1.75)
Bad school relationship (good)		1.20 (0.94~1.58)
Cox and Snell R^2^	0.16	0.18
Nagelkerke R^2^	0.23	0.24

Note. * *p* < 0.01, ** *p* < 0.001; OR—odds ratio; CI—confidence interval; ^a^ no suicidal ideation group as reference.

**Table 3 ijerph-18-10853-t003:** Factors associated with suicide plans.

Variables (Reference)	Suicide Plan (vs. Suicidal Ideation Only ^a^)	
	Model 1 Adjusted OR (95% CI)	Model 2 Adjusted OR (95% CI)
Male (female)	1.06 (0.83~1.35)	1.04 (0.82~1.33)
Age	1.12 * (1.03~1.22)	1.11 * (1.03~1.23)
Living with single or non-parent (living with parents)	1.37 ** (1.04~1.79)	1.33 ** (1.01~1.76)
Low subjective family class (not low)	1.04 (0.81~1.33)	0.93 (0.72~1.20)
Bullying victimization (no experience)	1.90 ** (1.50~2.40)	1.89 ** (1.49~2.41)
Depressive symptoms (no symptoms)	1.91 ** (1.47~2.50)	1.77 ** (1.34~2.32)
Anxiety (no anxiety)	1.47 ** (1.12~1.91)	1.42 ** (1.09~1.86)
Low subjective social support (not low)		1.26 (0.98~1.63)
Bad family relationship(good)		1.96 ** (1.46~2.63)
Bad school relationship(good)		0.75 (0.53~1.07)
Cox & Snell R^2^	0.07	0.09
Nagelkerke R^2^	0.11	0.13

Note. * *p* < 0.01, ** *p* < 0.001; OR = odds ratio; CI = confidence interval; ^a^ suicidal ideation only group as reference.

**Table 4 ijerph-18-10853-t004:** Factors Associated with Suicide Attempts.

Variables (Reference)	Suicide Attempt (vs. Suicide Plan ^a^)	
	Model 1 Adjusted OR (95% CI)	Model 2 Adjusted OR (95% CI)
Male (female)	1.01 (0.67~1.54)	1.01 (0.67~1.54)
Age	0.88 (0.76~1.02)	0.88 (0.76~1.02)
Living with single or non-parent (living with parents)	1.01 (0.70~1.72)	1.11 (0.70~1.74)
Low subjective family class (not low)	1.41 (0.93~2.14)	1.35 (0.85~2.26)
Bullying victimization (no experience)	0.83 (0.54~1.27)	1.09 (0.74~2.86)
Depressive symptoms (no symptoms)	1.26 (0.78~2.03)	1.39 (0.85~2.26)
Anxiety (no anxiety)	1.84 ** (1.15~2.97)	1.77 ** (1.10~2.87)
Low subjective social support (not low)		1.39 (0.91~2.12)
Bad family relationship(good)		1.14 (0.72~1.80)
Bad school relationship(good)		1.42 (0.82~2.51)
Cox & Snell R^2^	0.04	0.04
Nagelkerke R^2^	0.05	0.06

Note. ** *p* < 0.001; OR—odds ratio; CI—confidence interval; ^a^ suicide plan group as reference.

## Data Availability

The data that support the findings of this study are available from the corresponding author, upon reasonable request.

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
