# Peer review of "Influence of Experiencing Bullying Victimization on Suicidal Ideation and Behaviors in Korean Adolescents"

_ijerph, 2021, doi:10.3390/ijerph182010853_

Round 1

Reviewer 1 Report

It was pleasure to read this paper. Research topic is very actual  and important in today's society. Although the study has much potential, there are also several limitations that need to be addressed before the manuscript could be recommended for publication.

The Introduction part needs corrections. The text is a bit far from the title of the article. I suggest taking this into account. I would also suggest providing more information about the study problem - to present in more detail the latest research on bullying, phenomena that are associated with bullying and argue more strongly why you associate them with suicidal intention and how this varies depending on the age, gender, social status, etc. of the victims. Most likely, the specified object of research would help to better orient and reject unnecessary parts. 

Part Two - Materials and methods require adjustments as well. In the text, I would like to find a deeper explanation of why the data collected were chosen from 2015 years survey and why a specific age range was chosen for the analysis. It would also be good to get an explanation for why specific scales were chosen (such as anxiety, depression).

The results of the study are presented clearly.

Comparison of the obtained results with the researches of other authors would strengthen the part of the discussion.

The adjusted parts are likely to require adjustments to the findings part as well.

Author Response

Reviewer 1

Point 1.

The Introduction part needs corrections. I suggest taking this into account. I would also suggest providing more information about the study problem - to present in more detail the latest research on bullying, phenomena that are associated with bullying and argue more strongly why you associate them with suicidal intention and how this varies depending on the age, gender, social status, etc. of the victims. Most likely, the specified object of research would help to better orient and reject unnecessary parts. 

☞ Response: Thank you so much for your comments. We have added more information about relationship between bullying and suicide and other factors influencing their relationship (Line number: 62-83). And we have deleted unnecessary parts not related to the specific research questions.

Point 2.

Part Two - Materials and methods require adjustments as well. In the text, I would like to find a deeper explanation of why the data collected were chosen from 2015 years survey and why a specific age range was chosen for the analysis. It would also be good to get an explanation for why specific scales were chosen (such as anxiety, depression).

☞ Response: We did data from 2015 survey for this secondary analysis. Therefore, the target subjects, variables, and measurement tools for this study have already been selected and defined. We have added more explanation why we used the data from 2015 survey (Line 83-90) and why we used the specific variables such as depression and anxiety (Line 79-82). We also have added why we studied the adolescents aged 14 to 18 years (Line 63-64).

Point 3.

The results of the study are presented clearly. Comparison of the obtained results with the researches of other authors would strengthen the part of the discussion. The adjusted parts are likely to require adjustments to the findings part as well.

☞ Response: We have revised the discussion parts to add more other authors’ results (Line number: 287-313).

Reviewer 2 Report

In this manuscript, Kim and Ko investigate the relationship between bullying victimisation and suicidal ideation in an adolescent population. What's novel and particularly important about this study is that the authors have managed to pull apart the different stages leading up to suicide and investigate some factors that may contribute to these different stages. Overall, I believe this to be an important, relevant, and scientifically sound study, which should be published. 

I have very few minor comments I'd like to share with the authors:

  1. Page 2, lines 53-56, the discussion of the increase in psychological bullying due to changes in perceptions of physical violence is important to set the scene here. Thank you for including this.
  2. Page 2, lines 88-92, the wording of the second aim of this study is unclear (and throughout the manuscript as well). I would consider rewording to "factors that are associated with the development of suicidal ideation..."
  3. Page 8, line 280, I'd encourage the authors to consider that bullying isn't always associated with "violence" as this is typically physical and the introduction specifies a decrease in physical bullying and increase in psychological bullying.  

Author Response

Point 1.

Page 2, lines 88-92, the wording of the second aim of this study is unclear (and throughout the manuscript as well). I would consider rewording to "factors that are associated with the development of suicidal ideation..."

☞ Response: We used those terms such as ‘development of suicidal ideation’ and ‘transition from ideation to action’ based on the “Ideation-to-Action” Framework.

Reference:

  1. David Klonsky and Alexis M. May (2015). The three-step theory (3ST): a new theory of suicide rooted in the “Ideation-to-Action” Framework. International Journal of Cognitive Therapy, 8(2), 114-129.

Point 3.

Page 8, line 280, I'd encourage the authors to consider that bullying isn't always associated with "violence" as this is typically physical and the introduction specifies a decrease in physical bullying and increase in psychological bullying.  

☞ Response: We have added the discussion of type of bullying (Line 291-296)